# Admixture mapping identifies genetic regions associated with blood pressure phenotypes in African Americans

Zhi Liu[1], Daniel Shriner[2], Nancy F. Hansen[1], Charles N. Rotimi[2], James C. Mullikin[1,3]*, on behalf of the NISC Comparative Sequencing Program[3]¶

1 Cancer Genetics and Comparative Genomics Branch, National Human Genome Research Institute, National Institutes of Health, Bethesda, Maryland, United States of America, 2 Center for Research on Genomics and Global Health, National Human Genome Research Institute, National Institutes of Health, Bethesda, Maryland, United States of America, 3 NIH Intramural Sequencing Center, National Human Genome Research Institute, National Institutes of Health, Bethesda, Maryland, United States of America

¶ Author contributors of the NISC Comparative Sequencing Program are listed in the Acknowledgment section.
* mullikin@mail.nih.gov

## Abstract

Hypertension occurs at a higher rate in African Americans than in European Americans. Based on the assumption that causal variants are more frequently found on DNA segments inherited from the ancestral population with higher disease risk, we employed admixture mapping to identify genetic loci with excess local African ancestry associated with blood pressure. Chromosomal regions 1q21.2–21.3, 4p15.1, 19q12 and 20p13 were significantly associated with diastolic blood pressure ($\beta$ = 5.28, -7.94, -6.82 and 5.89, $P$-value = 6.39E-04, 2.07E-04, 6.56E-05 and 5.04E-04, respectively); 1q21.2–21.3 and 19q12 were also significantly associated with mean arterial pressure ($\beta$ = 5.86 and -6.40, $P$-value = 5.32E-04 and 6.37E-04, respectively). We further selected SNPs that had large allele frequency differences within these regions and tested their association with blood pressure. SNP rs4815428 was significantly associated with diastolic blood pressure after Bonferroni correction ($\beta$ = -2.42, $P$-value = 9.57E-04), and it partially explained the admixture mapping signal at 20p13. SNPs rs771205 ($\beta$ = -1.99, $P$-value = 3.37E-03), rs3126067, rs2184953 and rs58001094 (the latter three exhibit strong linkage disequilibrium, $\beta$ = -2.3, $P$-value = 1.4E-03) were identified to be significantly associated with mean arterial pressure, and together they fully explained the admixture signal at 1q21.2–21.3. Although no SNP at 4p15.1 showed large ancestral allele frequency differences in our dataset, we detected association at low-frequency African-specific variants that mapped predominantly to the gene *PCDH7*, which is most highly expressed in aorta. Our results suggest that these regions may harbor genetic variants that contribute to the different prevalence of hypertension.

**Data Availability Statement:** Data is available on dbGaP: https://www.ncbi.nlm.nih.gov/gap/advanced_search/?TERM=clinseq.

**Funding:** This research was supported by the Intramural Research Program of the National Human Genome Research Institute, National Institutes of Health, Bethesda, MD 20892.

**Competing interests:** The authors have declared that no competing interests exist.

## Introduction

Hypertension is a strong risk factor for coronary artery disease. Hypertension is a heritable disease, with an estimated heritability of 30–50% [1]. So far, genome-wide association studies (GWAS) have been successful at identifying genetic loci associated with hypertension, but the effect sizes of individual loci have been small, and currently-identified loci only explain a small fraction of the total heritability (4–6%) [2]. Other genetic determinants for blood pressure are still yet to be discovered.

One way to investigate the unexplained heritability is to study less represented populations, such as African Americans. In the United States, hypertension is diagnosed more frequently in African Americans than European Americans, representing 40% of African Americans vs. 28% of European Americans among people from all age groups [3], and 60% of African Americans vs. 38% of European Americans among individuals aged between 45 to 85 years [4]. African Americans form an admixed population in which each individual not only displays different proportions of European and African ancestry, but their chromosomes also show ancestral mosaicisms resulting from recombination across generations. To identify causal genetic loci in admixed populations, admixture mapping serves as a powerful tool when large allele frequency differences are present in the ancestral populations. Ancestral allele frequency differences at causal loci may contribute to disease prevalence differences in different populations.

Previously, GWAS for hypertension have identified genetic loci such as those containing mutations in genes *PDE3A* [5], *NOS3* [6] and *CYP17A1-CNNM2-NT5C2* [7]. In addition to GWAS, admixture mapping has also been successful at identifying ancestral haplotypes significantly associated with hypertension. Up until now, several loci have been reported to be associated with blood pressure, such as 6q24 and 21q21 [8]; within 21q21, *CXADR* was likely to play a role in blood pressure in African Americans [9]; by utilizing CARe consortium data, 5p13 was identified to be associated with diastolic blood pressure (DBP), with 3 uncorrelated SNPs within this region adequately accounting for the observed association [10]. For Hispanics, 6p12.3 was found to be associated with local African ancestry for mean arterial pressure (MAP) and DBP, but no variants were identified that drove these associations [11]. Compared to GWAS, admixture mapping has a much lower testing burden and thus requires much smaller sample sizes.

In this study, we utilized the ClinSeq® cohort exome sequencing data to identify genetic regions where local African ancestry was associated with blood pressure phenotypes. We fine-mapped these regions and identified genetic variants with large ancestral allele frequency differences that drove local ancestral associations. Furthermore, we replicated these variants identified from ClinSeq® in an independent cohort of Africans.

## Materials and methods

### Patient samples and exome-sequencing

The ClinSeq® study was approved by the Institutional Review Boards at the National Institutes of Health and informed consent was obtained from each participant. The ClinSeq® A2 cohort, which consists of 503 unrelated participants of self-reported African descent, aged between 45–65, were recruited between 2012–2017 and seen at the Clinical Center of the National Institutes of Health. Participants were not ascertained based on any particular phenotype and were interviewed and measured for various anthropometric and clinical variables. Blood samples were collected, from which DNA was isolated and exomic regions of interest were captured using the Integrated DNA Technologies (IDT) capture kit. Whole exome sequencing was

performed at the NIH Intramural Sequencing Center, Rockville, MD. Variant calling was performed using bam2mpg [12]. As a quality control (QC) step, single nucleotide polymorphisms (SNPs) were filtered for GQ $\geq$ 10 and GQ/DP $>$ 0.5; only autosomal SNPs were retained; SNPs discovered to be out of Hardy-Weinberg equilibrium (*P*-value $<$ 5.7E-07) based on an exact test [13], monomorphic SNPs, SNPs with call rate $<$ 0.98, and SNPs with minor allele frequency (MAF) $<$ 0.005 were removed. In a subsequent QC step, samples with mismatched sex, samples from related individuals and samples without phenotypes were removed. Based on principal component (PC) plots, samples outside of $\pm$ 4 standard deviations from the African American cluster by the first two eigenvalues were removed.

## Phenotypes

DBP and systolic blood pressure (SBP) were measured using a Dinamap instrument and obtained from the left arm after five minutes of rest with the subject in a sitting position with legs uncrossed. Two of these measurements were obtained and an average was taken for our analysis. For patients who took anti-hypertension medication, 5 mmHg and 10 mmHg were added to DBP and SBP, respectively [14]. MAP values were calculated as 1/3 $^*$ medically adjusted SBP + 2/3 $^*$ medically adjusted DBP. To check that values were normally distributed, we used a Shapiro-Wilk test. Untransformed DBP ($W$ = 0.99528, *P*-value = 0.1424), MAP ($W$ = 0.9942, *P*-value = 0.05881) and Log$_{10}$-transformed SBP ($W$ = 0.996, *P*-value = 0.2496) were found to be normally distributed (S1 Fig).

## Admixture mapping

Local ancestry was inferred using SEQMIX v0.12 [15]. For ancestry inference purposes, SNPs were pruned differently than in the association study. All markers, regardless of MAF, were retained. To ensure SNPs that were free of linkage disequilibrium (LD) with each other, SNPs were further pruned based on LD and sequencing depth: within a window size of 200, each step of 20, and r$^2$ threshold of 0.1, only one of a pair of two markers that had a higher sequencing depth was retained. After pruning, 316,761 markers were retained for local ancestry analyses. The CEU and TSI datasets from the 1000 Genome Project [16] were used as European population references; the YRI and LWK datasets were used as African population references. Global ancestry was inferred as principal components using LASER v2.04 (Locating Ancestry from SEquence Reads) (S2 Fig) [17]. The overall percentage of European and African ancestry for each individual was also estimated by averaging local ancestry across the individual's entire exome. Correlations were calculated between European, African ancestral percentage and principal components estimated by LASER (S3 Fig).

For admixture mapping, the following regression equation was used:

$$\text{Medically adjusted BP} = \beta_0 + \beta_1\,G_{AA} + \beta_2\,(G_{AA} - L_{AA}) + \beta_3\,\text{age} + \beta_4\,\text{age}^2 + \beta_5\,\text{sex} + \beta_6\,\text{BMI} + \varepsilon$$

$G_{AA}$ represents global African ancestry, and $L_{AA}$ represents local African ancestry at a specific locus. By definition, global ancestry and local ancestry are correlated. Therefore, we used the difference between global and local ancestry instead of just local ancestry for the purpose of easier interpretation of the resulting $\beta_2$. $\beta_2$ is interpreted as the local response accounting for the global ancestral effect, which every locus carries, and it predicts how the additional ancestry at each particular locus would contribute to the phenotype.

To estimate the effective number of tests, we used the method described by Shriner et al. [18]. The R package "coda" [19] was used for the estimation. More specifically, an autoregressive (AR) model was fitted to the vector of local African ancestry and the spectral density at

frequency zero was evaluated. The order of the fitted AR model is chosen by minimizing the Akaike Information Criterion (AIC). The effective number of tests for chromosomes for each individual was summed and then averaged across all individuals. In ClinSeq®, the total effective number of tests was equivalent to 66.84. Therefore, the genome-wide significant level $\alpha$ was 0.05/66.84 = 7.48E-04 (-$\log_{10} \alpha$ = 3.13), and significant regions were identified as contiguous regions within 1 unit drop of the peak LOD score, which yields approximately a 95% confidence interval.

## Regional association study

To identify SNPs that accounted for association signals discovered by admixture mapping, regional association studies were performed within admixture mapping significant regions. Since associated regions show a significant correlation between local ancestry and phenotype, we expect SNPs that drive the association signal to show a substantial allele frequency difference in different ancestral populations. Therefore, we only tested SNPs with allele frequency differences over 0.4 ($\delta > 0.4$) between European and African ancestral populations from the 1000 Genome Project. PLINK [20] was used to test the association between medically adjusted BP and genotypes at these SNPs, adjusting for the first 10 principal components to account for population structure. The following regression equation was used to perform the association analysis:

$$\text{Medically adjusted BP}$$
$$= \beta_0 + \beta_1 \, \text{genotype} + \beta_2 \, \text{age} + \beta_3 \, \text{age}^2 + \beta_4 \, \text{sex} + \beta_5 \, \text{BMI} + \sum_{i=1}^{10} \beta_{i+5} \, \text{PC}_i + \varepsilon$$

## Conditional admixture mapping

We performed conditional admixture mapping in order to estimate the degree to which associated SNPs explained the observed admixture mapping signals. For this step, the genotypes of significant SNPs from the regional association study were included as covariates in the admixture mapping equation to test whether one or more SNPs were able to account for admixture mapping signals. $P$-values for the local ancestry coefficient $\beta_2$ were recorded to investigate if they remained significant after adjusting for those SNPs. If $\beta_2$ was not significant after inclusion of a SNP, we interpreted it as the included SNPs were able to account for local ancestry effects observed in admixture mapping.

## Replication cohort description

We attempted to replicate our findings in the Africa America Diabetes Mellitus (AADM) cohort [21]. AADM is a study of type 2 diabetes in sub-Saharan Africans. The study is comprised of 5,231 participants recruited from university medical centers in Accra and Kumasi in Ghana; Enugu, Ibadan, and Lagos in Nigeria; and Eldoret in Kenya. Blood pressure was measured in the sitting position using an oscillometric device (Omron Healthcare, Kyoto, Japan). Three readings were taken with a ten-minute interval between readings. The reported DBP and SBP values were the average of the second and third readings. Weight was measured in light clothes on an electronic scale to the nearest 0.1 kg. Height was measured with a stadiometer to the nearest 0.1 cm. Body mass index was calculated as weight (kg) divided by the square of height ($m^2$). Individuals taking antihypertensive medication were excluded, leaving 2,957 individuals for analysis. For both DBP and SBP, values were inverse-normalized after adjusting for sex, age, and $age^2$. Genotyping was performed using the Affymetrix® Axiom® Genome-Wide PanAFR Array Set ($n = 1,808$) and the Illumina Infinium MEGA BeadChip, versions 1 ($n = 3,046$) and 2 ($n = 377$). For each array, quality control was performed as

described previously [22]. After excluding SNPs with a minor allele frequency < 5%, a genotyping call rate < 90%, or a Hardy-Weinberg $P$-value < 0.001, principal components analysis was performed on 124,266 SNPs common to all three genotyping arrays. For each array, imputation was performed using the African Genome Resources reference panel available from the Sanger Imputation Service, using EAGLE2 [23] for pre-phasing and PBWT [24] for imputation. Association testing was performed using a linear mixed model in EPACTS (https://github.com/statgen/EPACTS) [25], with body mass index and the first three principal components as fixed effects and the genetic relatedness matrix as a random effect. The reason for only adjusting three principal components is that according to the Tracy-Widom test [26], only three principal components were significant. The first PC separated Kenyans from Ghanaians and Nigerians and also separated the Kenyans. The second PC separated Ghanaians from Nigerians. The third PC separated 11 Yoruba. Additional PCs did not explain significant amounts of variance (S4 Fig). To account for the fact that AADM is enriched for cases of type 2 diabetes, we included type 2 diabetes status as a covariate in the association analysis. Ancestry proportions were inferred by projecting genotype data onto a previously described reference panel [27] using ADMIXTURE version 1.3.0 [28]. This study was approved by the Institutional Review Boards at each study site, Howard University, and the National Institutes of Health and informed consent was obtained from each participant.

## Results

### ClinSeq® cohort characteristics

484 individuals passed QC and were included in the final admixture mapping analyses. The cohort characteristics are shown in Table 1. The correlation between the average African ancestry and DBP, $\log_{10}$(SBP), and MAP were 0.121, 0.064 and 0.108, respectively (*P-value* = 0.007, 0.159 and 0.017, respectively), indicating that all three blood pressure phenotypes increased as the percentage of African ancestry increased.

### Local ancestry and global ancestry inference

The average estimated African ancestry in our study sample was 76.7±12.8%. The average number of ancestral switch points per individual was 127. European ancestry percentage was highly correlated with PC1, and African ancestry percentage was highly correlated with PC2, both with a correlation coefficient of r < -0.97 (S3 Fig).

### Admixture mapping

We next performed admixture mapping to identify regions where local African ancestry was significantly associated with blood pressure phenotypes. For DBP, four regions reached

**Table 1. Data description.**

| Characteristics | ClinSeq® | AADM |
|---|---|---|
| Age (SD) | 56 (6) | 50 (13) |
| Sex (Female %) | 74% | 63% |
| BMI (SD) | 32 (10) | 27 (6) |
| Diabetes | 17% | 50% |
| Anti-Hypertension Meds | 45% | 36% |
| Median African Ancestry (IQR) | 79% (16%) | 92% (13%) |
| Mean SBP (SD) | 125 (14) | 137 (24) |
| Mean DBP (SD) | 73 (9) | 82 (13) |

exome-wide significance: 1q21.2–21.3 (β = 5.28, $P$-value = 6.39E-04), 4p15.1 (β = -7.94, $P$-value = 2.07E-04), 19q12 (β = -6.82, $P$-value = 6.56E-05), and 20p13 (β = 5.89, $P$-value = 5.04E-04) (Fig 1A and Table 2). Among these four regions, two overlapped regions where local African ancestry was significantly associated with MAP: 1q21.2–21.3 (β = 5.86, $P$-value = 5.32E-04) and 19q12 (β = -6.40, $P$-value = 6.37E-04) (Fig 1B and Table 2). For SBP, no region reached exome-wide significance.

## Regional association test

Since SNPs with large allele frequency differences between ancestral populations carry the most information about ancestry, we identified SNPs that had over 40% allele frequency differences between European and African ancestral populations from the 1000 Genome Project within significantly associated regions. For DBP, 21 SNPs had δ > 0.4 and for MAP, 14 SNPs had δ > 0.4 (Table 2). These SNPs were tested for associations with DBP and MAP, respectively. Multiple testing thresholds were calculated using a Bonferroni correction. For DBP, the significance threshold was 0.05/21 = 0.00238 and for MAP, it was 0.05/14 = 0.00357. For DBP, SNP rs4815428, which is a 3' UTR variant downstream of the gene *TMC2*, reached significance after Bonferroni correction. For MAP, four SNPs reached significance after Bonferroni correction: rs3126067, rs2184953, rs58001094, and rs771205. SNPs rs3126067, rs2184953 and rs58001094 are in high linkage disequilibrium with each other and are all coding variants in *FLG*. SNP rs771205 is a missense variant in *MINDY1* (Table 3).

## Conditional admixture mapping test

After performing the regional association study, we tested whether significant SNPs can explain admixture mapping signals by including SNP genotypes as covariates in the admixture mapping model. If local African ancestry in the admixture mapping model fails to reach significance after adjusting for SNP genotypes, it indicates that the included SNPs can explain the admixture mapping signal. For DBP, only SNP rs4815428 was significant and passed the Bonferroni correction. By including rs4815428 as a covariate in the admixture mapping model, the peak $P$-value at chr20:2187986 increased from 5.04E-04 to 0.025. Local African ancestry failed to reach exome-wide significance. At the $P$-value < 0.05 level, the peak was still significant, indicating that there might be other markers weakly contributing to the admixture mapping signal (Fig 2A). For MAP, inclusion of rs3126067 increased the peak $P$-value from 5.32E-04 to 0.022 (Fig 2B). After adjusting for both rs3126067 and rs771205, the peak $P$-value increased to 0.146 (Fig 2B). This means the admixture mapping signal was completely explained by SNPs rs3126067 and rs771205 (or variants tagged by those two SNPs).

## Replication analyses

We attempted to replicate the association of five SNPs with blood pressure phenotypes in the AADM dataset of sub-Saharan African samples. No SNPs passed the genome-wide significant threshold used in AADM (S1 Table). We also performed an association study in the four identified admixture mapping peak regions (Table 4). In chromosome 4, the top SNP rs145765242 maps to gene *PCDH7*, which is most highly expressed in aorta according to GTEx [29]. The top SNP rs145765242 has an alternative allele frequency of 0.6% in the African samples in the Genome Aggregation Database (gnomAD) [30].

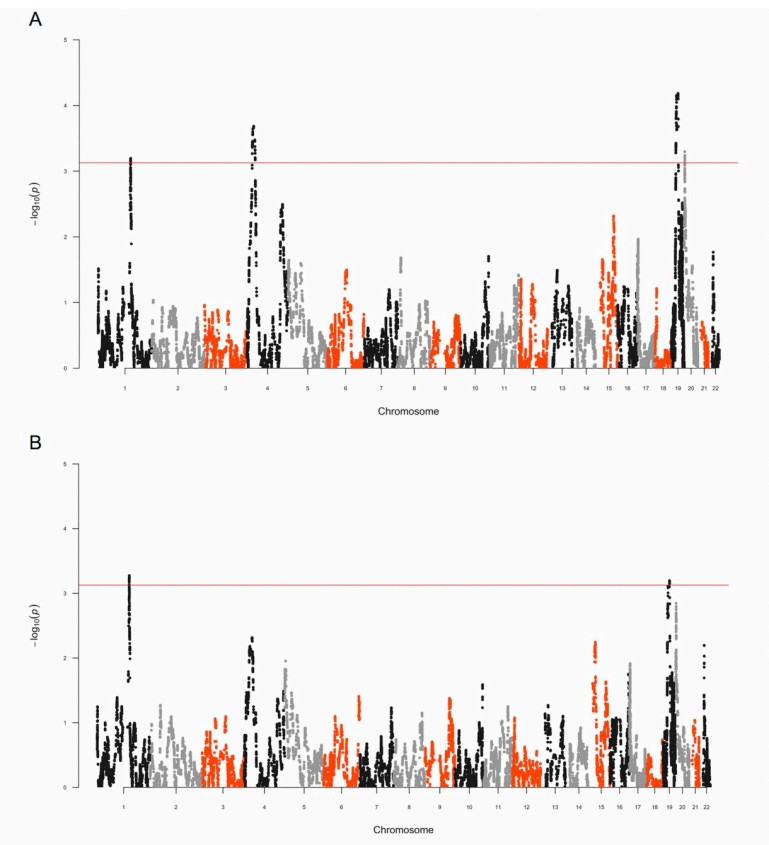

**Fig 1. Manhattan plot indicates chromosomal regions where local African ancestry is associated with DBP or MAP.** A) At chromosomal regions 1q21.2–21.3 (β = 5.28, *P*-value = 6.39E-04), 4p15.1 (β = -7.94, *P*-value = 2.07E-04), 19q12 (β = -6.82, *P*-value = 6.56E-05), and 20p13 (β = 5.89, *P*-value = 5.04E-04), local African ancestry was significantly associated with DBP. B) Two of the above regions overlapped regions where local African ancestry was significantly associated with MAP: 1q21.2–21.3 (β = 5.86, *P*-value = 5.32E-04) and 19q12 (β = -6.40, *P*-value = 6.37E-04). The red bar indicates the exome-wide significance threshold of 3.13.

## Discussion

We utilized admixture mapping methods to identify genetic regions associated with blood pressure phenotypes in African Americans. We identified four regions for diastolic blood pressure and two regions for mean arterial pressure that reached exome-wide significance in our admixture mapping study. Two MAP regions overlapped with the DBP regions, consistent with the fact that MAP is defined partially as a function of DBP.

**Table 2. Significant regions showing the association between blood pressure and African ancestry.**

| Chr | Region | Region (Mb) | Top signal location | Beta | Std. error | *P*-value | SNPs remained after QC | SNPs with δ > 0.4 |
|---|---|---|---|---|---|---|---|---|
| DBP | | | | | | | | |
| 1 | 1q21.2–21.3 | 148.2–154.7 | 1:151499346 | 5.28 | 1.54 | 6.39E-04 | 1034 | 16 |
| 4 | 4p15.1 | 30.7–31.1 | 4:31144153 | -7.94 | 2.12 | 2.07E-04 | 17 | 0 |
| 19 | 19q12 | 29.7–31.0 | 19:30020063 | -6.82 | 1.69 | 6.56E-05 | 84 | 0 |
| 20 | 20p13 | 0.6–5.3 | 20:2187986 | 5.89 | 1.68 | 5.04E-04 | 636 | 5 |
| MAP | | | | | | | | |
| 1 | 1q21.2–21.3 | 148.2–153.3 | 1:151501906 | 5.86 | 1.68 | 5.32E-04 | 782 | 13 |
| 19 | 19q12 | 29.7–32.1 | 19:30101339 | -6.40 | 1.86 | 6.37E-04 | 101 | 1 |

**Table 3. Significant SNPs within admixture mapping peak regions.**

| Chr | BP | rsID | Nearby genes | Std. error | Beta | A1 | A2 | A1 frequency | | δ | P-value[a] | Mutation type |
|---|---|---|---|---|---|---|---|---|---|---|---|---|
| | | | | | | | | EUR | AFR | | | |
| DBP | | | | | | | | | | | | |
| 20 | 2597978 | rs4815428 | *TMC2* | 0.73 | -2.42 | G | A | 0.34 | 0.85 | 0.51 | 9.57E-04 | Non-coding transcript exon |
| MAP | | | | | | | | | | | | |
| 1 | 150975108 | rs771205 | *MINDY1* | 0.67 | -1.99 | T | C | 0.03 | 0.68 | 0.65 | 3.37E-03 | Missense |
| 1 | 152276889 | **rs3126067** | *FLG* | 0.74 | -2.37 | G | A | 0.15 | 0.79 | 0.64 | 1.39E-03 | Synonymous |
| 1 | 152280782 | **rs2184953** | *FLG* | 0.73 | -2.33 | G | A | 0.18 | 0.79 | 0.61 | 1.44E-03 | Missense |
| 1 | 152283862 | **rs58001094** | *FLG* | 0.73 | -2.33 | C | G | 0.18 | 0.78 | 0.60 | 1.44E-03 | Missense |

[a] Multiple testing thresholds were calculated as 0.05/21 = 2.38E-03 for DBP and 0.05/14 = 3.57E-03 for MAP. Bold SNPs indicate they are in high linkage disequilibrium with each other ($r^2 > 0.99$, D' = 1, estimated in the dataset).

The significant admixture mapping region on chromosome 4 predominantly mapped to protocadherin 7 (*PCDH7*), which is most highly expressed in aorta according to GTEx [29]. Meta-analysis of nearly 35,000 individuals with African ancestry found SNP rs11931572, which tagged *PCDH7*, to be significantly associated with DBP [31]. The SNP rs11931572 has a low alternative allele frequency of 5% in Africans according to 1000 Genomes [16] and gnomAD [30]. This is consistent with our finding that this genetic region was significantly associated with African ancestry in both ClinSeq® and AADM datasets. However, utilizing admixture mapping drastically decreased the required sample size to identify this gene.

Within admixture mapping significant regions, five SNPs that had large ancestral allele frequency deviations (δ > 0.4) were significantly associated with BP. For DBP, SNP rs4815428 partially explained the admixture signal on chromosome 20. For MAP, SNPs rs3126067, rs2184953 and rs58001094 were in high LD with each other, and along with SNP rs771205, fully explained the admixture signal on chromosome 1.

SNPs rs3126067, rs2184953 and rs58001094 had previously been reported to be associated with ichthyosis and atopy [32] and atopic dermatitis [33]. All three SNPs are in the coding region of the gene filaggrin (*FLG)*, an intermediate filament-associated protein that aggregates keratin intermediate filaments in mammalian epidermis. Previously, this gene had been reported to be associated with asthma [34], ichthyosis [32] and abnormal inflammatory response [35, 36]. In a study of rheumatoid arthritis patients, first-degree relatives who were negative for rheumatoid arthritis but were positive for antibodies to citrullinated filaggrin had higher SBP and DBP than those who were antibody-negative [37], suggesting that there may be a correlation, although not necessarily a causation, between blood pressure and *FLG*.

SNP rs771205 is in the coding region of *MINDY1*, which encodes a hydrolase that removes lysine-48-linked conjugated ubiquitin from proteins [38]. It has exodeubiquitinase activity with a preference for long polyubiquitin chains and may play a regulatory role at the level of protein turnover [39]. This genetic region was previously reported to be linked to late-onset Alzheimer's disease [40, 41].

SNP rs4815428 is located in *TMC2*, which encodes transmembrane channel-like protein 2. *TMC2* is a potential ion channel required for the mechano-transduction of cochlear hair cells [42]. A study in the Han Chinese population reported that *TMC2* was among the top genes associated with BP response to the cold pressor test (CPT), which is associated with an increased risk of cardiovascular disease [43].

Previous admixture mapping studies of BP in African Americans were mostly performed on genotyping chip data with imputation, without much emphasis on whole exome

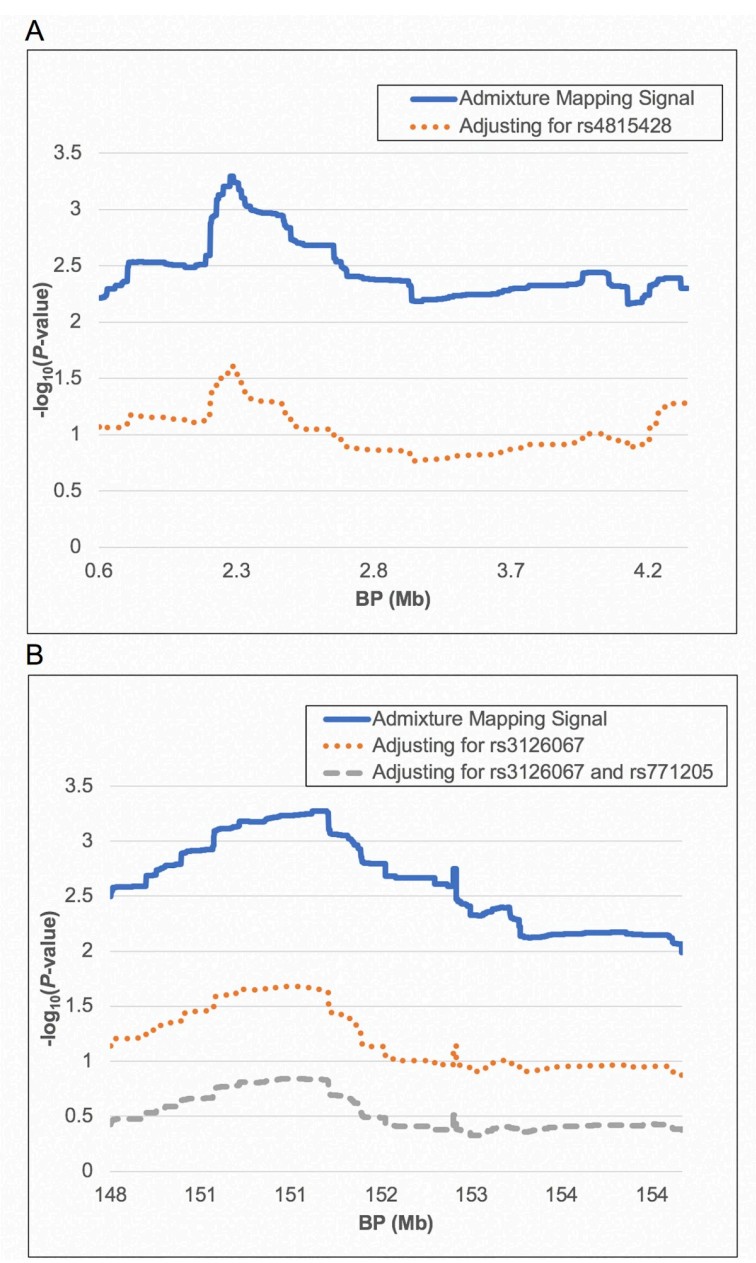

**Fig 2. Conditional admixture mapping study in DBP significant region chr20:0.6–5.3 Mb and MAP significant region chr1:148.2–153.3 Mb.** A) For DBP in the region of chr20:0.6–5.3 Mb, after adjusted for SNP rs4815428, the significant signal partially disappeared. SNP rs4815428 partially explained the significant admixture mapping signal in this region. B) For MAP in the region of chr1:148.2–153.3 Mb, after adjusted for SNPs rs3126067 and rs771205, the significant signal completely disappeared. These two SNPs fully explained the significant admixture mapping signal in this region. The x-axis is not to scale due to missing intronic regions.

sequencing (WES) data. WES data provide greater coverage of rarer variants at exonic regions than chip data, therefore enabling us to test more variants for local ancestry association signals. Although the five SNPs we identified were all common SNPs, it is also possible for rare variants to drive the admixture mapping signals as well. With 484 African American samples, we could not identify any rare variants that passed multiple testing correction (results not shown). In

**Table 4. Top associated SNPs within admixture mapping peak regions for DBP in the AADM dataset.**

| Chr | BP | REF | ALT | rsID | Nearby gene | Alt. freq | Stat | *P*-value | Beta | Std. error | $R^2$ | Alt. freq$_{AFR}$ | Alt. freq$_{EUR}$ |
|-----|----|----|----|------|-------------|-----------|------|-----------|------|-----------|-------|----------|----------|
| 1 | 150579702 | TA | T | rs11360645 | *ENSA* | 0.198 | -3.63 | 2.84E-04 | -0.12 | 0.03 | 0.0045 | 0.192 | 0.549 |
| 4 | 30813550 | C | A | rs145765242 | *PCDH7* | 0.00947 | 3.77 | 1.65E-04 | 0.48 | 0.13 | 0.0048 | 0.00794 | 0 |
| 19 | 30950681 | C | T | rs150339768 | *ZNF536* | 0.00118 | -4.01 | 6.27E-05 | -1.45 | 0.36 | 0.0054 | 0.00099 | 0.00398 |
| 20 | 3200824 | CT | C | NA | *ITPA* | 0.425 | -4.10 | 4.31E-05 | -0.10 | 0.03 | 0.0057 | NA | NA |

addition, utilizing WES data may leave out ancestral switch points occurring within non-exonic regions, causing the average number of switch points to be lower than those estimated using WGS or chip data.

Our five significantly-associated SNPs were not replicable in the AADM study, possibly due to the following reasons. 1) The African American individuals recruited in the ClinSeq® study were relatively healthy individuals, mostly without an identifiable disease trait such as diabetes or cardiovascular disease, whereas AADM was enriched for cases of type 2 diabetes. However, the inclusion of type 2 diabetes status as a covariate made negligible differences in the association statistics. 2) The African ancestry in the African Americans ClinSeq® study might not be sufficiently matched to the Africans in the AADM study. Participants were recruited for the AADM study from three countries: Nigeria, Ghana, and Kenya. Individuals from Nigeria and Ghana are expected to share ancestry with African Americans, based on historical records of the trans-Atlantic slave trade. However, the contribution to African Americans from other places such as Senegal is not as well captured by AADM. 3) The AADM participants were recruited on the African continent, while the ClinSeq® participants were recruited in North America. Differences in environmental factors between the studies may have contributed to relatively smaller effect sizes in AADM than in ClinSeq®. 4) The causal variants may be specific to European ancestry. 5) The SNPs associated in ClinSeq® are not the SNPs underlying differential risk in the regions identified by admixture mapping.

To demonstrate that these five SNPs are not just ancestry-informative markers (i.e., confounders), we stratified samples based on their local ancestry at each SNP and performed genotypic association within each stratum, which, by definition, cannot be confounded by local ancestry. We then combined all strata and performed a random effects meta-analysis to get an overall estimation of the association for each SNP. We were able to demonstrate that genotypic association at rs4815428 was not confounded by local ancestry. This SNP has the largest effect size among all five SNPs. We were not able to demonstrate that the rest of the four SNPs were not confounded (S2 Table). After stratification, each stratum has a very small sample size. We believe that increasing the study sample size may give a more definitive answer.

In summary, we performed admixture mapping analyses on the ClinSeq® African American cohort and identified four genetic regions associated with blood pressure phenotypes. We fine-mapped these regions and identified five SNPs that are the main driving forces of associations between local African ancestry and blood pressure phenotypes at two of the four regions. SNP rs3126067, rs2184953 and rs58001094, all located in *FLG*, and SNP rs771205, located in *MINDY1*, were significantly associated with MAP. SNP rs4815428, located in *TMC2*, was significantly associated with DBP. Finally, region 4p15.1, despite containing no SNPs exhibiting large allele frequency deviation in our dataset, maps predominantly to *PCDH7*, which is most highly expressed in aorta.

## Supporting information

**S1 Fig. Distribution of systolic blood pressure, diastolic blood pressure, and mean arterial pressure for the ClinSeq® A2 dataset.** The phenotype distributions are approximately normal.
(TIFF)

**S2 Fig. Scatter plot for ClinSeq® A2 dataset in HGDP panel.** This figure shows principal components 1 and 2 of the ClinSeq® A2 dataset using the HGDP reference panel. The red cluster represents African ancestral populations; the blue cluster represents European ancestral populations; the yellow cluster represents East Asian populations, and the purple cluster represents Native American populations from the HGDP reference panel. The black cluster represents the ClinSeq® study.
(TIFF)

**S3 Fig. Correlation between global European ancestry and principal component 1 and global African ancestry and principal component 2.** This figure shows the global European ancestry estimated by averaging local ancestry across the entire exome plotted against principal component 1 as estimated by LASER, and the global African ancestry plotted against principal component 2. The X axis denotes global ancestry; the Y axis denotes principal components. Each dot denotes an African American individual who has passed QC.
(TIFF)

**S4 Fig. Principal component analysis plots of 5,231 participants from the AADM Study.** Principal component 1 separates Kenyans from Ghanaians and Nigerians and also separates the Kenyans. Principal component 2 separates Ghanaians from Nigerians. Principal component 3 separates 11 Yoruba.
(TIFF)

**S1 Table. Replication study in the AADM dataset.**
(DOCX)

**S2 Table. Meta-analyses of local-ancestry stratified genotypic association tests.**
(DOCX)

## Acknowledgments

We sincerely thank the patients and their families for their participation and support of this project. We thank the medical staffs for taking clinical measurements and patient samples collection. We thank all the authors for their work towards completion of this project. We specially thank Dr. Leslie Biesecker for his initiation of the ClinSeq® project, reviewing and commenting on the manuscript.

NISC Comparative Sequencing Program author contributors: Beatrice B. Barnabas, MPH, MSc; Sean Black, MSc; Gerard G. Bouffard, PhD; Shelise Y. Brooks, BS; Holly Coleman, MSc; Lyudmila Dekhtyar, MSc; Xiaobin Guan, PhD; Joel Han, BS; Shi-ling Ho, BS; Richelle Legaspi, MSc; Quino L. Maduro, BS; Catherine A. Masiello, MSc; Jennifer C. McDowell, PhD; Casandra Montemayor, MSc; Morgan Park, PhD; Nancy L. Riebow, BS; Karen Schandler, MSc; Chanthra Scharer, BS; Brian Schmidt, BS; Christina Sison, BS; Sirintorn Stantripop, BS; James W. Thomas, PhD; Pamela J. Thomas, PhD; Meghana Vemulapalli, MSc; Alice C. Young, BA. The lead author for NISC Comparative Sequencing Program: James C. Mullikin, PhD. Email: mullikin@mail.nih.gov.

## Author Contributions

**Conceptualization:** Zhi Liu.

**Formal analysis:** Zhi Liu.

**Funding acquisition:** James C. Mullikin.

**Methodology:** Daniel Shriner.

**Supervision:** Nancy F. Hansen, Charles N. Rotimi, James C. Mullikin.

**Validation:** Daniel Shriner.

**Writing – original draft:** Zhi Liu.

**Writing – review & editing:** Daniel Shriner, Nancy F. Hansen, Charles N. Rotimi, James C. Mullikin.

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
