## [Decision Letter · Decision Letter 0]

28 Jan 2020

PONE-D-19-36060

Admixture mapping identifies genetic regions associated with blood pressure phenotypes in African Americans

PLOS ONE

Dear Dr. Mullikin,

Thank you for submitting your manuscript to PLOS ONE. After careful consideration, we feel that it has merit but does not fully meet PLOS ONE’s publication criteria as it currently stands. Therefore, we invite you to submit a revised version of the manuscript that addresses the points raised during the review process.

We would appreciate receiving your revised manuscript by Mar 13 2020 11:59PM. To enhance the reproducibility of your results, we recommend that if applicable you deposit your laboratory protocols in protocols.io, where a protocol can be assigned its own identifier (DOI) such that it can be cited independently in the future. For instructions see: http://journals.plos.org/plosone/s/submission-guidelines#loc-laboratory-protocols

We look forward to receiving your revised manuscript.

Kind regards,

Heming Wang, PhD

Academic Editor

PLOS ONE

Additional Editor Comments (if provided):

In general this manuscript is technically sound and well-written. One key question raised by reviewer 1 is lack of replication. Can you try to look up those variants in African-American exome array data such as FBPP, ARIC, or MESA? Since you mentioned that those variants may reflect European ancestry association, replication using European cohorts may also be interesting.

Journal Requirements:

2. One of the noted authors is a group or consortium [NISC Comparative Sequencing Program]. In addition to naming the author group, please list the individual authors and affiliations within this group in the acknowledgments section of your manuscript. Please also indicate clearly a lead author for this group along with a contact email address.

Reviewers' comments:

Reviewer's Responses to Questions

**Comments to the Author**

1. Is the manuscript technically sound, and do the data support the conclusions?

Reviewer #1: Partly

Reviewer #2: Yes

2. Has the statistical analysis been performed appropriately and rigorously? 

Reviewer #1: Yes

Reviewer #2: Yes

3. Have the authors made all data underlying the findings in their manuscript fully available?

Reviewer #1: Yes

Reviewer #2: Yes

4. Is the manuscript presented in an intelligible fashion and written in standard English?

Reviewer #1: Yes

Reviewer #2: Yes

5. Review Comments to the Author

Reviewer #1: This is a well-written manuscript focusing on identifying genetic loci associated with blood pressure traits in African Americans using admixture mapping.

Major concern/suggestion:

Overall, the discovery analyses are adequate but the authors have failed to show whether the replication cohort would be a suitable sample for replication analysis. ClinSeq participants were African Americans who were not ascertained for any particular phenotype but the AADM participants were Africans with T2D. I'd like the authors to explain the rationale behind using AADM as a replication cohort instead of another African American population-based study, with similar estimated African ancestry. I would strongly suggest incorporating additional replication analysis considering how poorly AADM replicated the discovery findings. In Table 1, please update the sample characteristics to include the replication cohort(s) so that readers can easily examine the similarities and differences between the discovery and replication samples.

Minor questions/suggestions:

-Page 9: The authors stated "For both DBP and SBP, values were inverse-normalized after adjusting for sex, age, and age^2." Other studies have included BMI as a covariate in this step, what was the reason to omit BMI in this instance?

-Page 10: What was the reason for only adjusting 3 PCs for AADM?

-Page 10: Please add median & SD of SBP and DBP in Table 1.

Reviewer #2: The paper addresses an important topic related to genetic susceptibility to hypertension in African Americans, who have a high burden of hypertension compared to other ethnic/racial groups in the US. Genome-wide association studies (GWAS) have identified a small number of blood pressure loci in individuals of African ancestry, with some challenges being sample size required for association studies and the ancestral admixture in this population. In this paper, the authors used an approach that leverage the European and African admixture (which have been shown to identify novel loci not uncover in GWAS) and exome sequencing data from a cohort of African Americans to identify genetic regions where local African ancestry was associated with hypertension-related phenotypes. Using this approach, they identified four genomic regions where local African ancestry was significantly associated with DBP, with two of them overlapping with significant findings for MAP. Some interesting findings are related to subsequent association analyses within these regions that identified SNPs that partially or completely account for the admixture mapping findings in conditional analyses. The authors attempted to replicate findings in an African study of diabetes, although results were not significant. This is a well-though and carefully done study with multiple strengths in the design, approach and follow-up analyses. The authors are well-known leaders in the field. I have some minor comments to help clarify some of the aspects of the study.

For the admixture mapping, models used the difference between global and local ancestry instead of just local ancestry. This may be of interest to highlight in the paper and perhaps include the reason for this approach.

Page 11, please clarify what you mean by average number of ancestral switch points per individuals was 127.

Table 1, add the interquartiles for African ancestry %, so one can see the range of the African admixture in the discovery sample.

It may be of interest to include the interpretation of betas for African local ancestry shown in Table 2.

Table 3. For the SNPs in LD (chromosome 1), include in the footnote id the LD was estimated in the data (or 1000G Project reference panel).

6. PLOS authors have the option to publish the peer review history of their article (what does this mean?). If published, this will include your full peer review and any attached files.

Reviewer #1: No

Reviewer #2: No

---

## [Author Response · Author response to Decision Letter 0]

13 Mar 2020

Please see attached PDF "Response to Reviewers.pdf" to see figures in the response. However, here is the text of the response:

March 10th, 2020

Dear Reviewers,

We sincerely thank you for your time and efforts on reviewing this manuscript. We have addressed all comments in a revised version of the manuscript. Please see our responses to each specific comment below.

1. Additional Editor Comments (if provided):

In general this manuscript is technically sound and well-written. One key question raised by reviewer 1 is lack of replication. Can you try to look up those variants in African-American exome array data such as FBPP, ARIC, or MESA? Since you mentioned that those variants may reflect European ancestry association, replication using European cohorts may also be interesting.

ARIC and MESA are prospective studies of atherosclerosis in middle-aged adults. These studies are not population-based but rather are ascertained for individuals asymptomatic for cardio-vascular disease. Hence, participants in these studies are expected to not have hypertension at baseline. The FBPP is the parent program of GENOA, which is a cohort study with ascertainment for sibships in which at least two siblings had essential hypertension diagnosed prior to 60 years of age. Like ARIC and MESA, this study is not population-based. Thus, these studies would not satisfy Reviewer 1.

2. Reviewer #1: This is a well-written manuscript focusing on identifying genetic loci associated with blood pressure traits in African Americans using admixture mapping.

Overall, the discovery analyses are adequate but the authors have failed to show whether the replication cohort would be a suitable sample for replication analysis. ClinSeq participants were African Americans who were not ascertained for any particular phenotype but the AADM participants were Africans with T2D. I'd like the authors to explain the rationale behind using AADM as a replication cohort instead of another African American population-based study, with similar estimated African ancestry. I would strongly suggest incorporating additional replication analysis considering how poorly AADM replicated the discovery findings. In Table 1, please update the sample characteristics to include the replication cohort(s) so that readers can easily examine the similarities and differences between the discovery and replication samples.

We are unclear what the reviewer’s basis was for claiming that AADM “poorly replicated the discovery findings” and we do not agree that “considering how poorly AADM replicated the discovery findings” necessarily leads to the conclusion that non-replication is a failure and hence that AADM is an unsuitable replication study. It is plausible that AADM is performing as expected to rule out variants that do not explain the admixture signal. A genetic epidemiology study of West Africans that reflects descendants of either the parental African populations or populations genetically close to the parental African populations, which AADM does (and is currently the only such study), is suitable for replicating associations of specific variants. Even if we use another African American population with similar estimated mean percentage of African ancestry, that population could represent different parental ancestries of African and European combinations. To account for the fact that AADM is enriched for cases of type 2 diabetes, we included T2D status as a covariate in the association analysis (p. 10). This change is shown in Table 4 and Table S1.

For a rare European-specific variant (minor allele frequency less than 1%), in a typical African American population (e.g., 20% European and 80% African ancestry), the chance to find it in regions of European ancestry in African Americans is about 20% x 1%, which is close to 0%. This drastically reduces the power we have to replicate rare variants in African Americans, so we opted to focus on common or African-specific variants and used an African replication cohort.

We updated the sample characteristics and included the replication cohort in Table 1.

2.1. Page 9: The authors stated "For both DBP and SBP, values were inverse-normalized after adjusting for sex, age, and age^2." Other studies have included BMI as a covariate in this step, what was the reason to omit BMI in this instance?

We included BMI as a covariate during the association analysis using EPACTS. Not adjusting for BMI at the earlier step facilitated analyses of different models using EPACTS, i.e., including or excluding BMI as a covariate, without having to generate multiple versions of the blood pressure phenotypes.

2.2. Page 10: What was the reason for only adjusting 3 PCs for AADM?

The Tracy-Widom test (https://www.ncbi.nlm.nih.gov/pubmed/22441298) revealed three significant PCs. The first PC separates Kenyans from Ghanaians and Nigerians and also separates the Kenyans. The second PC separates Ghanaians from Nigerians. The third PC separates 11 Yoruba. Additional PCs do not explain significant amounts of variance. This explanation is added to the “Replication cohort description” section.

PCA plots of 5,231 participants from the AADM Study

2.3. Page 10: Please add median & SD of SBP and DBP in Table 1.

The mean and standard deviation of SBP and DBP are added in Table 1.

3. Reviewer #2: The paper addresses an important topic related to genetic susceptibility to hypertension in African Americans, who have a high burden of hypertension compared to other ethnic/racial groups in the US. Genome-wide association studies (GWAS) have identified a small number of blood pressure loci in individuals of African ancestry, with some challenges being sample size required for association studies and the ancestral admixture in this population. In this paper, the authors used an approach that leverage the European and African admixture (which have been shown to identify novel loci not uncover in GWAS) and exome sequencing data from a cohort of African Americans to identify genetic regions where local African ancestry was associated with hypertension-related phenotypes. Using this approach, they identified four genomic regions where local African ancestry was significantly associated with DBP, with two of them overlapping with significant findings for MAP. Some interesting findings are related to subsequent association analyses within these regions that identified SNPs that partially or completely account for the admixture mapping findings in conditional analyses. The authors attempted to replicate findings in an African study of diabetes, although results were not significant. This is a well-though and carefully done study with multiple strengths in the design, approach and follow-up analyses. The authors are well-known leaders in the field. I have some minor comments to help clarify some of the aspects of the study.

3.1. For the admixture mapping, models used the difference between global and local ancestry instead of just local ancestry. This may be of interest to highlight in the paper and perhaps include the reason for this approach.

We used the difference between global and local ancestry instead of just local ancestry for the purpose of easier interpretation of the resulting beta. Global ancestry is the average of local ancestry across all loci; consequently, global and local ancestry are correlated. Our beta is interpreted as the local response accounting for the global ancestral effect, which every locus carries, i.e., it predicts how the additional ancestry at each particular locus would contribute to the phenotype. We highlighted this in the fourth paragraph of the admixture mapping method section.

3.2. Page 11, please clarify what you mean by average number of ancestral switch points per individuals was 127.

Individual 1, Chromosome 1

In the illustrated figure above, for individual 1, chromosome 1, the ancestry switched five times for the first haplotype and two times for the second haplotype, so the total number of switch points for individual 1, chromosome 1 is 5+2=7.

We calculated the total times of ancestry switching from African to European or vice versa for each chromosome of each individual, and then added that up for all chromosomes of each individual. We added the total switch points for all chromosome of all individuals and divided by the total number of individuals to get the average number of ancestral switch points per person as 127.

The denser the markers used to detect ancestry, the more ancestry switches are likely to be detected. We used exome sequencing data, which does not target intergenic regions. Off-target reads may not be adequate to identify all ancestry switches in such regions. Therefore, our average switch points are lower than those identified using whole genome sequencing or genotyping data.

3.3. Table 1, add the interquartiles for African ancestry %, so one can see the range of the African admixture in the discovery sample.

The median and interquartile ranges of the percent African ancestry for both the discovery and replication studies have been added to Table 1.

3.4. It may be of interest to include the interpretation of betas for African local ancestry shown in Table 2.

Beta can be interpreted as how much can excess local African ancestry contribute to an individual’s blood pressure at that particular locus.

3.5. Table 3. For the SNPs in LD (chromosome 1), include in the footnote id the LD was estimated in the data (or 1000G Project reference panel).

The LD level is R2 > 0.99 and D’ = 1 for any pair of the three SNPs, which was estimated from the dataset. This was indicated in Table 3.

We would like to thank you again for reviewing our work. If you have any questions or concerns regarding this manuscript, please feel free to address them to me at mullikin@mail.nih.gov. Thank you for your time and consideration of this manuscript.

Sincerely,

James C. Mullikin

Head, Comparative Genomics Unit

Director, NIH Intramural Sequencing Center

National Human Genome Research Institute 

National Institutes of Health 

5625 Fishers Lane 

Room 5N-01Q 

Rockville, MD 20852 

Tel (301) 496-2416, Fax (301) 435-6170

---

## [Decision Letter · Decision Letter 1]

7 Apr 2020

Admixture mapping identifies genetic regions associated with blood pressure phenotypes in African Americans

PONE-D-19-36060R1

Dear Dr. Mullikin,

We are pleased to inform you that your manuscript has been judged scientifically suitable for publication and will be formally accepted for publication once it complies with all outstanding technical requirements.

With kind regards,

Heming Wang, PhD

Academic Editor

PLOS ONE

Additional Editor Comments (optional):

The authors have sufficiently addressed the reviewers' questions. I understand their challenge in acquiring additional replication data and eager to publish this paper as soon as possible. Just to clear, ARIC and MESA are considered as population-based studies.

Reviewers' comments:

Reviewer's Responses to Questions

**Comments to the Author**

1. If the authors have adequately addressed your comments raised in a previous round of review and you feel that this manuscript is now acceptable for publication, you may indicate that here to bypass the “Comments to the Author” section, enter your conflict of interest statement in the “Confidential to Editor” section, and submit your "Accept" recommendation.

Reviewer #1: (No Response)

Reviewer #2: All comments have been addressed

2. Is the manuscript technically sound, and do the data support the conclusions?

Reviewer #1: Yes

Reviewer #2: Yes

3. Has the statistical analysis been performed appropriately and rigorously? 

Reviewer #1: Yes

Reviewer #2: Yes

4. Have the authors made all data underlying the findings in their manuscript fully available?

Reviewer #1: Yes

Reviewer #2: Yes

5. Is the manuscript presented in an intelligible fashion and written in standard English?

Reviewer #1: Yes

Reviewer #2: Yes

6. Review Comments to the Author

Reviewer #1: I'd like to thank the authors for addressing my comments and suggestions in the revised manuscript. The only part that I am still unclear about is the rationale behind the authors stating that AADM is the only study that "reflects descendants of either the parental African populations or populations genetically close to the parental African populations". There is no indication within the manuscript that ClinSeq participants are descendants of West Africans (and if it's the case, then it should be specified). If the discovery cohort is a population-based cohort not ascertained for any specific phenotypes/diseases, then have the authors considered using an African American biobank cohort for additional replication?

Reviewer #2: The authors have answered appropriately all my questions and I have no further suggestions for this paper.

7. PLOS authors have the option to publish the peer review history of their article (what does this mean?). If published, this will include your full peer review and any attached files.

Reviewer #1: No

Reviewer #2: No

---

## [Editor Report · Acceptance letter]

9 Apr 2020

PONE-D-19-36060R1 

Admixture mapping identifies genetic regions associated with blood pressure phenotypes in African Americans 

Dear Dr. Mullikin:

I am pleased to inform you that your manuscript has been deemed suitable for publication in PLOS ONE. Congratulations! Your manuscript is now with our production department. 

With kind regards,

on behalf of

Dr. Heming Wang 

Academic Editor

PLOS ONE